# Investigation of the Exposure of Schoolchildren to Ultrafine Particles (PM_0.1_) during the COVID-19 Pandemic in a Medium-Sized City in Indonesia

**DOI:** 10.3390/ijerph20042947

**Published:** 2023-02-08

**Authors:** Rizki Andre Handika, Worradorn Phairuang, Muhammad Amin, Adyati Pradini Yudison, Febri Juita Anggraini, Mitsuhiko Hata, Masami Furuuchi

**Affiliations:** 1Graduate School of Natural Science and Technology, Kanazawa University, Kanazawa 920-1192, Japan; 2Faculty of Science and Technology, Jambi University, Jambi 36364, Indonesia; 3Faculty of Geosciences and Civil Engineering, Institute of Science and Engineering, Kanazawa University, Kanazawa 920-1192, Japan; 4Air and Waste Management Research Group, Faculty of Civil and Environmental Engineering, Institut Teknologi Bandung, Bandung 40132, Indonesia; 5Faculty of Environmental Management, Prince of Songkla University, Hat Yai 90110, Thailand

**Keywords:** school environments, personal exposure, ultrafine particles, PM_0.1_, questionnaire survey

## Abstract

The health risk of schoolchildren who were exposed to airborne fine and ultrafine particles (PM_0.1_) during the COVID-19 pandemic in the Jambi City (a medium-sized city in Sumatra Island), Indonesia was examined. A questionnaire survey was used to collect information on schoolchildren from selected schools and involved information on personal profiles; living conditions; daily activities and health status. Size-segregated ambient particulate matter (PM) in school environments was collected over a period of 24 h on weekdays and the weekend. The personal exposure of PM of eight selected schoolchildren from five schools was evaluated for a 12-h period during the daytime using a personal air sampler for PM_0.1_ particles. The schoolchildren spent their time mostly indoors (~88%), while the remaining ~12% was spent in traveling and outdoor activities. The average exposure level was 1.5~7.6 times higher than the outdoor level and it was particularly high for the PM_0.1_ fraction (4.8~7.6 times). Cooking was shown to be a key parameter that explains such a large increase in the exposure level. The PM_0.1_ had the largest total respiratory deposition doses (RDDs), particularly during light exercise. The high level of PM_0.1_ exposure by indoor sources potentially associated with health risks was shown to be important.

## 1. Introduction

In modern society, air pollution is one the most challenging problems since it can cause severe problems not only for the environment but also for human health. This is particularly true for fine particulate matter (PM_2.5_), which are particles with an aerodynamic diameter of less than 2.5 µm [1,2]. Previous epidemiological and toxicological studies on PM_2.5_ have reported a wide range of adverse health effects related to the cardiovascular and respiratory systems, even causing premature mortality [3,4]. Since it is well established that PM_2.5_ is one of the leading environmental risk factors in the global burden of disease [5] and that the smaller the particle the more harmful its effect to human health, the focus of scientific research in recent years has also begun to shift to tiny particles such as submicron (<1 µm) and ultrafine particles (<0.1 µm or UFPs) sizes [6].

UFPs, also referred to as airborne nanoparticles or PM_0.1_, are generated from engineering and combustion processes largely from vehicles, biomass burning, etc. Previous studies confirm that PM_0.1_ is very harmful not only due to its size but also due to its unique physicochemical characteristics [7,8,9,10,11,12,13,14]. PM_0.1_ can penetrate deeply into the alveolar region, then reach the bloodstream before finally being translocated to sensitive areas of the human body. These particles can also cause oxidative stress, chronic/acute inflammatory disorders or cancer, even at low concentrations [14,15,16], particularly in children [17,18]. Since children’s respiratory and immune systems are still developing, and their respiratory rate is higher than adults, this leads to a higher possible exposure to UFPs [19,20,21]. Due to the short and size-dependent lifetime of UFPs, the distribution of these particles in the atmosphere is extremely nonhomogeneous. Some studies have reported that, the higher the altitude, the higher the number concentration, which makes estimating the concentration levels rather difficult particularly for estimating health effects on humans [11,22]. Furthermore, personal exposure in the breathing zone (PBZ ~30 cm hemisphere around the mouth and nose) allows PM_0.1_ exposure to be better and more accurately evaluated with respect to actual particle concentration levels and in every microenvironment to which people are exposed. Hence, monitoring and characterizing data on UFPs have become critical, particularly in the case of personal exposure for preparation to achieving “a new normal” of post-COVID-19.

Previous investigations on children’s daily personal exposure to UFPs before the COVID-19 pandemic have been intensively conducted in European and Australian cities/countries [23,24,25,26], while it has also been studied in Bhutan [27], Ghana [28] and China [29]. These studies have identified the home microenvironment to be the most significant contributor to children’s daily exposure to UFPs, with many factors related, but it is mainly due to cooking activities [23,24,27,28,29]. In Southeast Asia (SEA) cities/countries, just one pilot study currently has been conducted [30], but it has still not given sufficient information about daily exposure to UFPs, whereas the ambient PM_0.1_ levels in most SEA countries were found to be much worse than in Western countries [31], confirming the necessity to extend investigations in the form of personal exposure. In addition, information concerning exposure to UFPs based on the critical sources and exposure status during pandemic conditions are needed; in Indonesia, schoolchildren were allowed back to school at the end of 2021 after spending almost one year online.

Although many previous studies have emphasized the exposure to UFPs in terms of number and surface area [10,11], information on the actual mass of UFPs or PM_0.1_ remains limited due to the limitation of air sampling tools for PM_0.1_. On the other hand, the mass concentration is essential for comprehensively understanding the characteristics of UFPs linked not only to doses and mass, but also chemical components. Studies on mass base exposure to size-segregated particles down to the PM_0.25_ level have largely been conducted on elderly people in the indoor home [32], mail carriers while delivering mail outdoors [33], and the indoors and outdoors of residential homes [34]. However, personal exposure to size-segregated particles down to the PM_0.1_ level is still scarce, particularly regarding schoolchildren during the COVID-19 pandemic.

Within this context, the objective of this study was to evaluate the exposure to size-segregated particles down to PM_0.1_ by targeting junior high school students who live in different urban settings (urban and suburban) in the medium-sized city of Jambi, Indonesia. Their health risks are also discussed through their respiratory deposition doses based on personal PM exposure data. A questionnaire survey was used to collect information on schoolchildren from selected schools and included collecting information such as personal profiles, living conditions, daily activities and health status. Size-segregated ambient particles at selected locations in school environments (five schools) were collected for a 24-h period, while the personal exposure of PM to 40 selected schoolchildren from those schools was evaluated during the daytime using a personal air sampler capable of collecting PM_0.1_ particles.

## 2. Materials and Methods

### 2.1. Site Locations and Characteristics

Five public schools in Jambi City, located in Sumatra Island, Indonesia, were selected as target sites for the present study (Figure 1). Jambi City is the capital city of Jambi Province, with a total population of 606,200 people in an area of 205.4 km^2^ [35]. As shown by the authors [36], the air pollution in the urban area of Jambi City is strongly affected by road vehicles. In addition, from July to October, peatland fires become a key factor concerning air pollution since Jambi City is surrounded by large peatland areas [37]. It also has a significant impact on cross-border pollution in other countries, including Singapore, Malaysia, Southern Thailand, Brunei and the Philippines [38,39,40]. This, however, drastically dropped during the COVID-19 pandemic in the past two years [41].

In Jambi City, there were 26,998 schoolchildren aged 12 to 15, who were educated in 73 junior high schools, which included 26 public and 47 private institutions [42]. Thus, five public schools located on prominent roadsides (RS) with a larger student population than private schools were chosen; three in the urban area (U1, U2 and U3) and two in the suburban area (SU1 and SU2) (Figure 1). Medium-sized cities in Indonesia are generally inter-connected with other cities so that there is a large difference in traffic characteristics between urban areas (the central part of a city) and suburban areas (the outer areas of the city). This may cause a large difference in micro-environments from the PM pollution point of view. Since contributions of each micro-environment experienced by participants to personal exposure is a key issue of this study, different school environments were selected based on different categories, i.e., a central city area, an urban area and a suburban area.

### 2.2. Questionnaire for Schoolchildren’s Characteristics and Behaviors

A questionnaire survey was conducted to collect information on schoolchildren from the selected schools and included personal information and daily activities via Google Forms (https://forms.gle/DK31RnwKfLcmposV7, accessed on 23 March 2022). Living conditions, daily activities and health status were then analyzed. The questionnaire was administered to a total of 719 children (U1: 12.2%, U2: 21.7%, U3: 33.1%, SU1: 21.7%, and SU2: 11.3%) aged 12–15 years, consisting of 276 males (38.4%) and 443 females (61.6%), where the female ratio was slightly larger than that of schoolchildren overall (~50%). Health outcomes other than respiratory symptoms, e.g., via spirometry tests, were not evaluated in the present study in order to have minimum contact with participants and others under the COVID-19 pandemic conditions.

### 2.3. Air Sampling in School Environments

Size-segregated ambient PM at selected locations in school environments were collected for 24 h. At two locations, namely, the school gate (SG) and schoolyard (SY), the sampling was conducted simultaneously and repeated three times on weekdays and once on the weekend. In urban school 1 (U1), sampling was conducted two times on weekdays and once on the weekend. Information on the protocol is summarized in Table 1.

A cascade impactor that was devised using inertial filter technology [43] was used for the air sampling. The air sampler is herein referred as “Ambient Nano-Sampler (ANS)” (Figure 2a). The ANS consists of four impactor stages (PM_>10_, PM_10–2.5_, PM_2.5–1.0_, PM_1.0–0.5_), an inertial filter (IF) stage (PM_0.5–0.1_) and a backup filter (PM_<0.1_) located downstream of the inertial filter stage [44]. It was operated at an airflow rate of 40 L min^−1^. Two sets of ANS were simultaneously run for 24 h at the SG and SY. Quartz fibrous filters (QFF) (2500 QAT-UP, Pall Corp., New York, USA) with a diameter of 55 mm were used in all impactor and backup filter stages after a pre-treatment procedure described below. The IF stage consisted of a cartridge with a circular nozzle of Ø 5.25 mm containing webbed stainless-steel fibers (fiber diameter, *d_f_* = 9.8 μm, Nippon Seisen Co. Ltd., Osaka, Japan, felt type, SUS-316) of designed total weight that was packed uniformly.

### 2.4. Evaluation of Personal Exposure

The personal exposure of schoolchildren to PM was evaluated using a personal air sampler capable of collecting PM_0.1_ particles. The personal exposure near the breathing zone of each participant was evaluated for 12 h during the daytime. The measurements were designed to start at 7:30 in the morning before the participants traveled to each school and were continued until 7:30 p.m. Eight participants were selected from each school (total 8 × 5 = 40 participants). The participants also answered survey questions while providing written informed consent. They were requested to behave normally and to fill out the time activity diary (TAD) for each 15-min time slot (e.g., 07:00–07:15, 07:15–07:30, 07:30–07:45, etc.).

The participants who were able to complete the measurements numbered 34 and comprised nine boys and 25 girls, i.e., five (U1), eight (U2), six (U3), eight (SU1), and seven (SU2) (details in Appendix A). Four students could not complete the sampling because the portable pump had stopped functioning before finishing the sampling, and two others failed for some other reason.

A personal air sampler shown in Figure 2b was used for the exposure measurements. The Personal Nano-Sampler (PNS) was developed and revised by Furuuchi et al. (2010) and Thongyen et al. (2015) [45,46] and is applicable for the evaluation of PM_0.1_ exposure. The PNS consists of two impactor stages (PM_10 or 2.5/1_) and two IF stages (PM_0.4–1_, PM_0.1_) that are configured in line, and a backup filter located downstream of PM_0.1_ IF. The first stage of the pre-cut impactor was covered by silicon grease (Dow Corning, 03253589) to a uniform thickness of around 0.2 mm during each sampling. Teflon-bound glass fiber (TBF) filters (TX40HI20-WW Pall Corp., New York, NY, USA) with diameters of 10 and 47 mm were attached on the second pre-cut impactor (PM_1–2.5_) and used as a backup filter. The pre-cut inertial filter consists of webbed SUS fibers (the same fibers that are used for ANS) packed in a circular nozzle. The main inertial filter consists of five-layer mesh TEM grids (Silver mesh G600HSS, 600 mesh, mesh width *d_f_* = 5 μm, pitch = 42 μm, mesh thickness *t* = 8 μm) sandwiched by five spacers with a circular hole diameter of 1.9 mm (spacer thickness = 30 μm) in an aluminum cartridge [46], and it was used for collecting PM_0.1–0.4_. A portable battery pump (ASP-6000, KOMYO RIKAGAKU KOGYO K.K., Kawasaki, Japan) was connected to the outlet of the PNS unit by a flexible resin tube. The sampling air flow rate was 4.0 L min^−1^.

### 2.5. Filter Preparation and Weighing

The QFF filters were pre-baked for 1 h at 350 °C to remove any possible contamination following the guidelines of The Ministry of Environment, Japan [47] for the future chemical analysis. The QFF and TBF filters and IF cartridges were both stored in a weighing chamber (PWS-PM2.5, Tokyo Dylec Corp., Tokyo, Japan) for 48 h at a controlled temperature of 21.5 ± 1.5 °C and relative humidity of 35 ± 5%. The filters were then weighed before and after sampling using an electric microbalance (Sartorius Cubis MSU2.7S-000-DF, minimum digit (MD) = 1 μg) installed inside the weighing chamber.

### 2.6. Quality Assurance and Quality Control

The pump flow rate was calibrated using a HORIBA VP-4U bubble flow meter. Filters for sampling were accompanied by travel blanks to account for possible contamination during the sampling and transportation. Each filter was placed into a plastic (polyethylene) bag while covered by aluminium foil to avoid any chemical contamination. Each mass of filter samples was subtracted from the mean value of travel blank filter. The sample concentrations below the blank value were excluded from the data. The minimum detection limit (3σ) based on the evaluation of travel blanks was 5 µg (47-mm TBF filter), 2 µg (10-mm TBF filter), and 115 µg (55-mm QFF filter), respectively. These values were somewhat lower than the minimal value of filter samples, or 17 µg (47-mm TBF filter), 4 µg (10-mm TBF filter), and 119 µg (55-mm QFF filter), respectively.

Before the sampling, the participants received a lecture in which the study design including objectives and items for reliable measurements, etc. were outlined to ensure that the data collected for personal exposure were acceptable. The detailed methodology was also provided concerning how to record the details of the TAD and how to wear the PNS. Before the sampling, the participants were requested to learn how to operate the PNS by themselves. In order to immediately respond to any problems during the measurements, the SNS through mobile phones was used for the communication between the surveyors and the participants. Pump batteries were charged after each measurement, and a new measurement was then started after checking for equipment malfunctions.

### 2.7. Estimation of Respiratory Deposition Doses by Personal Exposure 

The respiratory deposition doses (RDDs) were also estimated to determine the total deposition in the respiratory tracts for each size fraction of particles (PM_<0.1_, PM_0.1–0.4_, PM_0.4–1.0_ and PM_1.0–2.5_) during a day (±12 h). According to the International Commission on Radiological Protection (ICRP), RDDs can be evaluated by Equation (S1) [48]. All particles were assumed to penetrate completely via the nose or mouth into the respiratory tract. The particle deposition fraction (*DF*) was calculated in three regions of the respiratory tract, i.e., the head airways (*DF_HD_*) (Equation (S2)) with the inhalable fraction (*IF*) calculated by Equation (S3), tracheobronchial (*DF_TB_*) (Equation (S4)) and alveolar (*DF_AL_*) (Equation (S5)), to determine the RDD values. The ICRP model (1994) [49] has often been used in previous studies for adults, generally in light exercise and seated conditions. The calculation of RDDs for schoolchildren was first used in this study based on previous studies concerning particle deposition in children. Children might have a slightly larger *DF* [19,20] and have a greater ventilation rate per body weight or lung surface area, which may result in different tissue burdens compared with adults [21]. However, these differences only affected the mouthpiece [50] for increasing tidal volume compared with normal relaxed breathing. Thus, Bennet & Zeman (1998) [50] concluded that the deposition fraction of 2-µm particles is not different in children aged 7–14 years compared with adults, particularly during resting with spontaneous breathing, while a clear difference in *DF* from adult values was found for children younger than 9 [51]. In addition, the relative contributions to their tidal volume regarding the rib cage and abdomen by the posture of children were essentially the same as that in adults [52]. Person gender and physical activity status determined the values of a tidal volume or *VT* (m^3^ breathe^–1^) and the typical breath frequency or *f* (breath min^–1^) [53,54,55]. Therefore, the RDD values were calculated separately for the exposure of male and female students. Two different exposure conditions were assumed, i.e., light exercise and seating. In light exercise, the *VT* and *f* values were assumed to be 9.9 × 10^−4^ (12.5 × 10^−4^) m^3^ breath^−1^ and 21 (20) breath min^−1^ for males and females, respectively. In the seated position, 4.6 × 10^−4^ (7.5 × 10^−4^) m^3^ breath^−1^ and 14 (12) breath min^−1^ were used for females and males, respectively [48].

### 2.8. Statistical Analysis

The normal distribution status of the quantitative data was first checked before applying a statistical analysis, using the Shapiro–Wilk test for 7 ≤ number of samples (N) ≤ 50 and the Lilliefors test for 5 ≤ N ≤ 6 [56]. Parametric and non-parametric tests were used to analyze the data by ANS, PNS, and schoolchildren’s characteristics, respectively (Appendix A). A ρ-value of less than 0.05 was also considered for all the performed tests.

## 3. Results and Discussion

### 3.1. General Characteristics of Schoolchildren

As seen from Appendix A, most of the schoolchildren had no severe respiratory problems (68.0%), while 10% or less had respiratory symptoms, i.e., a common cold, a cough, a cough and cold, and breathlessness. Of these students, 13.5% had suffered from COVID-19. During the survey period, on-site schooling in the classroom was allowed for ~4 h per day (~16%) and the schoolchildren wore masks during these periods. In addition, school canteens were still not operational and schoolchildren who had symptoms of COVID-19 were not allowed to come to school. Their parents took them to school mostly using motorcycles (61.5%) or cars (29.5%), while a few of the schoolchildren used public transportation (2.7%) or walked (6.3%). The traveling time between home and school (round-trip) was less than 2.0 h (~4%). Hence, schoolchildren spent their time mostly on indoor activities (19 to 22 h, ~88%) while between 2 and 5 h per day (~12%) were spent on traveling and outdoor activities.

The schoolchildren lived together with their parents and siblings in a house. Most of the residents lived in housing estates that were located far from the roadside (±85.4%) but these locations were surrounded by small roads. Residential occupant numbers were 3–4 (70.5%), 5–8 (25.7%) and >8 (3.8%). Their homes were generally ventilated by the natural wind so that air exchange between outdoors and indoors was not very efficient, particularly for fine particles that were generated indoors [54,57].

Indoor PM emission is associated with human activities, and it contributes significantly to the production of fine and ultrafine particles in indoor air [58,59]. In 60.2% of the surveyed homes, smoke caused by both cigarette smoking and cooking was observed. According to the survey, cooking generally lasted for 2–3 h per day (±91.9%) but some cooking required 4–5 h and >5 h (7.0% and 1.1%), respectively. In most cases, the schoolchildren participated in cooking twice a day; in the morning before going to school around 8–9 A.M., and in the afternoon after coming back from school around 12–1 P.M. The average cooking durations in the morning and afternoon were 1.4 ± 0.6 and 2.3 ± 1.3 h, respectively. Although fewer male children participated in cooking, they spent most of their time in a family room or in a dining room that was in very close proximity to the kitchen. Hence, it is possible that some of the male children were also exposed to cooking activities.

### 3.2. Mass Concentrations of Ambient PM in School Outdoor Environments

In Table 2, the mass concentration of each size fraction of PM in the school outdoor environments is summarized, and the mass ratio between different PM sizes is shown in Figure 3. The average PM mass concentrations including PM_0.1_ as well as the peak mass percentage for the PM_2.5–10_ fraction are consistent with reported observations in other cities in Southeast Asia [60,61,62]. Average PM mass concentrations at the school gate (SG) were larger than those at the schoolyard (SY) for all size ranges and site locations, particularly in the case of the coarsest fraction (>10 µm). However, the difference between SG and SY was larger at suburban sites while the PM distribution was rather uniform at the urban sites. Such tendencies were possibly related, not only to the degree of influence of road traffic, but also to emissions that occurred in the schoolyards as well as those in school surroundings. Since the canteens in schools, the likely sources in the schoolyard area, were closed during the pandemic period and the activities of schoolchildren were restricted to inside the yard, the most likely anthropogenic sources may have been from traffic emissions. The larger difference between SG and SY at the suburban site, therefore, may be attributed to a lower background value in the area and a larger influence by heavier traffic in a neighboring road. In Table 3, the present results are compared with those from a similar case of roadside and schoolyard environments in Medan City in North Sumatra [63] along with roadside values from other cities [36,64]. In the case of Medan City, the PM concentration was quite large because of the much larger amount of traffic, and it increased in the schoolyard probably because, in the survey year (2019), the canteen was normally operated and there were no restrictions on schoolchildren’s activities in the schoolyard. For a detailed and rigorous discussion on the source apportionment, a chemical component analysis is needed. However, this issue will be addressed in the near future. From the results shown here, we conclude that the school outdoor environment was a key area where the schoolchildren stayed for ~4 h per day, and that the air quality was not affected by some unique sources other than that of traffic but, on average, it was similar in both areas and typical of values for other cities in this area.

### 3.3. Schoolchildren’s Personal Exposure

#### 3.3.1. Mass-Size Fractions of PM Exposure

Figure 4 shows the average mass concentration of size-segregated particles down to PM_0.1_ that the schoolchildren were exposed to for each school. Although, because of individual differences, the actual average duration of exposure measurement was 11.8 ± 0.5 h, the average concentration value may be acceptably regarded as the 12-h time-weighted average (12-h TWA). The exposure to PM_0.1_ and PM_0.4–1_ appeared to be dominant-size fractions in most cases, in which the PM_0.1_ accounted for around 40% of PM_≤2.5_, i.e., fine particles, with an average PM_0.1_ exposure of 25.9 ± 10.1 µg m^−3^. The schoolchildren in suburban areas were exposed to PM_0.1_ more heavily, exceeding 50 µg m^−3^ for some children from the SU2. These high levels of PM_0.1_ are comparable to ambient PM levels during remarkable episodes such as in Chiang Mai city, during the forest fire periods in 2014–2015 in northern Thailand [65] and during the peatland fire season in 2019 in Sumatra, Indonesia [36].

The exposure of the schoolchildren to all PM size fractions was statistically compared considering the schoolchildren’s school origin and area where they lived (urban or suburban). Concerning the results of normal distribution tests, the ANOVA test was used for the analysis of personal exposure to PM_0.1_, PM_0.1–0.4_, and PM_0.4–1_. Meanwhile, for the exposure to the PM_1–2.5_ fraction, the Kruskal–Wallis and the Mann–Whitney tests were used, respectively, to evaluate the differences between all schoolchildren based on their school, and between schoolchildren in urban and suburban areas. These results are summarized in Table 4. Large differences in PM_0.1_ and PM_0.1–0.4_ with the origin of the school are noted (F > F_critical_, the ANOVA test), while a noticeable difference between areas appeared only in the case of the PM_0.1_ fraction. From the results listed in Table 5, which were obtained through the post hoc test using α_Bonferroni-corrected_, the largest difference between schools for the PM_0.1_ was found between U3 (17.0 ± 7.5 µg m^−3^) and SU1 (29.5 ± 5.7 µg m^−3^), while the largest difference in PM_0.1–0.4_ exposure was between SUI (15.0 ± 7.3 µg m^−3^) and SU2 (5.6 ± 4.2 µg m^−3^).

From the above results, it became clear that the exposure level of PM_0.1_ was quite high and that there was a large difference in the exposure level for the submicron fractions (<1 µm), particularly the PM_0.1_. Such results can be attributed to the dominant emission sources in each environment where participants lived and also to the amount of PM exposure to the specific activities such as residence time and frequency of opportunities for exposure.

#### 3.3.2. Personal vs. Ambient School Environments of Size-Resolved PM

In Figure 5, the average PM levels for personal exposure (PE) and outdoor school environments are compared for PM_0.1_, PM_1_ and PM_2.5_. The average PE levels are also summarized in Table 6 in comparison with the average outdoor PM levels. The PE level was 1.5~7.6 times higher than the outdoor PM level particularly for the PM_0.1_ fraction (4.8~7.6 times). The ratio of PE to outdoor levels, PE/SY and PE/SG, had peaks at PM_0.1_ both for the urban and suburban groups. Since personal exposure depends on behaviors of the participants, such factors that can result in higher levels of PM_0.1_ exposure need to be analyzed based on the time of activity and the microenvironments of the schoolchildren during the personal exposure measurements. According to the results of the questionnaire survey, indoor and outdoor activities (including transit time), respectively, shared ~88% and ~12%; the large ratios of PE to SY and SG suggest that schoolchildren had opportunities to be exposed to more contaminated microenvironments (MEs) outside school that contained more fine and ultrafine fractions of PMs.

### 3.4. Schoolchildren’s Exposure Due to Time-Activity and Microenvironments

During the personal exposure measurements, the participating schoolchildren also reported the times of their activities and the corresponding microenvironments (MEs) (Appendix A). Most of their activities were at home; in the living room, kitchen/dining room, bedroom, and other rooms in the home (total ~7 h). The transit or commuting time was classified as “go to school,” “back from school”, “go to others” and “back from others” (total ~1 h). “In-classroom”, “outside classroom”, “others (indoor)” and “others (outdoor)” were further categorized to describe differences in school environments (~4 h). These three different categories of environments—during transit, in the school environment and at home—are discussed below as key factors that can attribute to the exposure to PMs.

#### 3.4.1. Schoolchildren’s Exposure during Transit and during School Environments

Previous studies using personal online PM sensors concluded that commuting environments provided the highest opportunity for exposure, particularly to UFPs, and that it occurred in a very short time period per day [23,29]. In the present cases, the schoolchildren’s commuting activity was mostly between home and school accompanied by parents for about 1 h, either using a motorcycle or a car. Such a difference appears to be reasonable because of the direct exposure to a traffic environment during transit by motorcycle. Although the contribution of commuting between the school environment and personal exposure PM levels was not so large, it would appear that on-road exposure is an important factor.

On the other hand, exposure in a school environment contributed less than 10% to the total exposure while schools were fully operative [24,28]. In this study, the influence of the school environment on schoolchildren’s exposure tended to be lower than in normal situations since school activity had just started again after nearly one year of online instruction and accounted for 4 h per day, and school canteens, as the most possible source in the yard area, were still closed. As discussed above, personal exposure levels were 1.5~7.6 times higher than the outdoor PM levels, particularly for PM_0.1_ (4.8~7.6 times).

#### 3.4.2. Influences of Indoor Home Sources of PM on Personal Exposure

Since it was indicated that activities while schoolchildren were at home were the most important for exposure to PMs, their characteristics in each ME at home and PM concentrations during personal sampling were analyzed. As possible parameters related to indoor sources, the number of occupants, time period of cooking per day and smoking status at home were evaluated, as in the following (Table 7).

The average exposure level with smokers at home increased by 8~20% from that without smokers, with the largest increase in the suburban group (29.5% for the average PM_2.5_). This suggests that home smoking has a certain level of influence. However, the findings indicated that it was not so significant probably because children were normally not close to smoking areas at home. Meanwhile, the exposure level increased nearly linearly with the time cooking, as shown in Figure 6a, in which the average PM concentration of each size category are plotted against the average cooking period for male and female children. As seen from Figure 6b, the PM_0.1_/PM_2.5_ ratio also increased linearly with the time of cooking. The maximum PM_0.1_ concentration and the ratio PM_0.1_/PM_2.5_ became around five and three times those for 1 to 5 h of cooking time, respectively. Although each individual average value was less reliable for the cooking time <1 h (n = 2), the mass concentration and fraction of PM_0.1_ for 1 h were quite similar to those of outdoor environments. The other categories of PM tended to be similar but less sensitive to the cooking period. Although there was a slight difference in tendency between male and female children, the data were not sufficient to allow the differences to be specified. Regarding the number of home occupants, the influence was not clear, with differences of 0.96~0.99 times between 3–4 and 5–8 occupants.

From these results, we conclude that the exposure to indoor PM emissions caused by cooking is the most important key parameter that determines the much higher exposure level of PM compared with the outdoor PM concentration. For further discussion, the chemical component analysis will be needed to clarify the contribution of cooking and smoking at home and traffic during commuting.

### 3.5. Estimated Respiratory Deposition Doses of Each Size-Fraction of Particles

Respiratory deposition doses (RDDs) of schoolchildren calculated using Equation (S1) are summarized in Figure 7 and Table 8. For both categories of activities, the largest total RDDs appeared in the PM_<0.1_ fraction followed by PM_0.4–1_ and PM_1–2.5_, indicating the importance of dosing by fine fraction particles with sizes less than 1 μm. The RDDs during light exercise were larger than those for the seated position, and these were mostly attributed to male students. Such an increase in the RDDs under conditions of light exercise were also reported for PM_2.5–10_ and PM_1.0–2.5_ by Kumar & Jain (2021) [54] and Segalin et al. (2017) [32]. This may be explained by an increase in breathing frequency during the light exercise [48]. The larger RDDs for male children can be explained by a larger tidal volume for intake that can increase the deposition of PM in the respiratory tract [32,53]. It should be noted that the largest differences in RDDs between males and females appeared for the PM_0.1_ fraction (5.9%) and, as seen from Figure 7, such quite large RDDs for PM_0.1_ were mainly from the deposition of particles in the alveolar region. Since PM_0.1_ particles contain large amounts of hazardous chemicals per unit particle mass [12,66], possible adverse health influences would be predicted. For a detailed and rigorous discussion concerning the deposition doses to children based on their time-activity and the internal dose of metals (As, Pb, Mn, Cd, Cr) in the human body (e.g., kidney), an extended analysis, such as the use of ExDoM2 [67,68,69], would be needed. However, this issue will be addressed in the near future for schoolchildren’s personal PM_0.1_ exposure.

## 4. Conclusions

The health risk for schoolchildren was discussed based on the mass concentration of exposed airborne fine and ultrafine particles (PM_0.1_) and schoolchildren’s behaviors in their daily life during a period of the COVID-19 pandemic in Jambi City, a medium-sized city in Sumatra Island, Indonesia. The 12-h average personal exposure level in schoolchildren was evaluated to be 1.5~7.6 times higher than that for the outdoor level measured in school environments and it was particularly high for the PM_0.1_ fraction (4.8~7.6 times). The schoolchildren spent most of their time in indoor (~88%) environments with only ~12% for transit (~1 h) and outdoor activities including schooling (~4 h), suggesting that the indoor environment is the dominant contributor to schoolchildren’s exposure. As one piece of evidence for the contribution of indoor emission, the exposure to indoor PM emission by cooking was found to be the most important parameter that describes the much higher exposure level of PM than the outdoor PM concentrations. Such a contribution appeared to be proportional to the period of cooking at home, particularly for the PM_0.1_ mass concentration and a mass fraction of PM_0.1_ to PM_2.5_. Although there was an influence of surrounding circumstances of targeted schools, such as more influence of heavy vehicle traffic in suburban areas, it contributed only slightly to the increased exposure. Home smoking and a larger exposure experienced by motorcycle commuting also increased the exposure but still had much less influence than cooking. The largest total RDDs of PM_0.1_, revealed that, particularly during light exercise, particles infiltrated deeper into the alveoli. Overall, the study highlighted the significance of personal PM_0.1_ exposure, which was identified as more varied based on location than other size-resolved PM. The high levels of PM_0.1_ exposure in the present study demonstrated the larger importance of sources that are potentially associated with health risks, particularly concerning cooking at home. The findings of this study will aid in improving schoolchildren’s health, particularly with respect to the PM_0.1_ fraction. This study can also serve as a guide for future directions in assessing population exposure and developing air pollution mitigation strategies such as the better management of ventilation in housing in Indonesia, especially concerning children’s health risks. This is important during such a period of social restriction under the COVID-19 pandemic.

## Figures and Tables

**Figure 1 ijerph-20-02947-f001:**
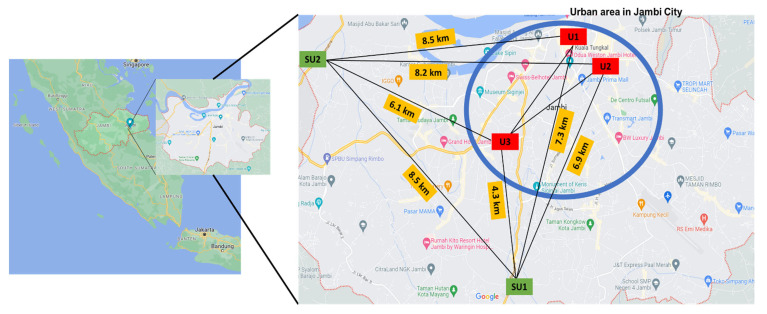
Study sites and selected schools (U1, U2, U3, SU1, and SU2) in Jambi city, Indonesia.

**Figure 2 ijerph-20-02947-f002:**
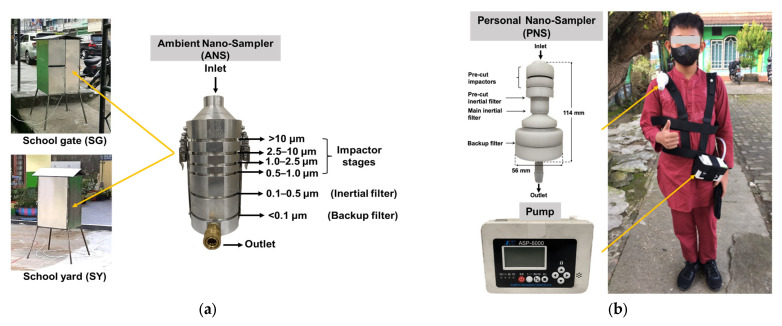
(**a**) Ambient Nano-Sampler (ANS) and (**b**) Personal Nano-Sampler (PNS) and description of use in field samplings.

**Figure 3 ijerph-20-02947-f003:**
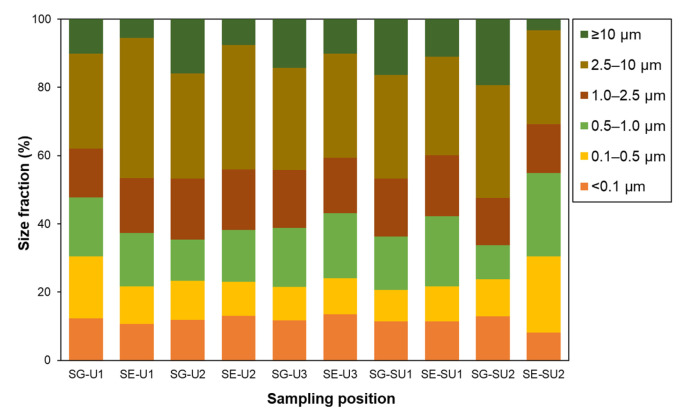
Average percentage of size fractions (particle size: <0.1 µm, 0.1–0.5 µm, 0.5–1.0 µm, 1.0–2.5 µm, 2.5–10 µm, and ≥10 µm) to total particles (TSP) at SG and SE in each school.

**Figure 4 ijerph-20-02947-f004:**
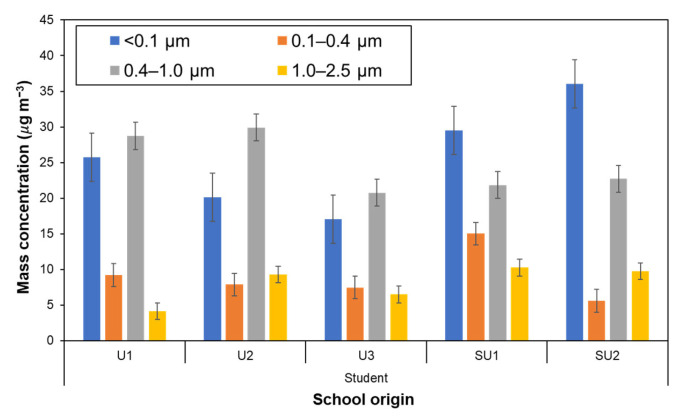
Average mass concentrations of size-segregated particles down to the PM_0.1_ fraction to schoolchildren participants by personal exposure.

**Figure 5 ijerph-20-02947-f005:**
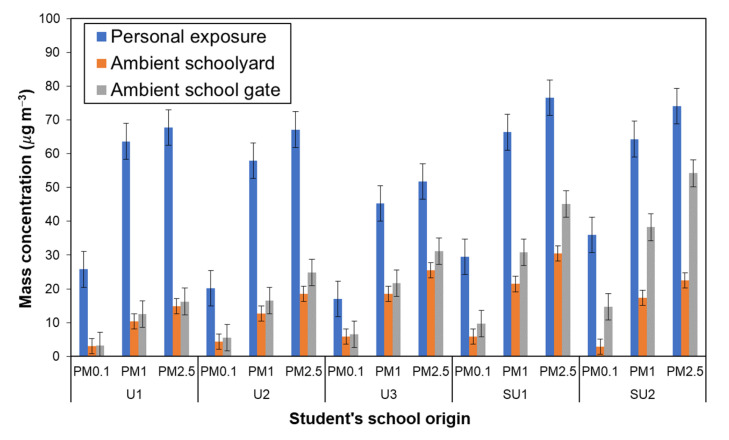
Size-resolved particle (PM_0.1_, PM_1_, and PM_2.5_) concentrations based on schoolchildren’s school of origin: personal exposure vs. in ambient schoolyard (SY) and school gate (SG).

**Figure 6 ijerph-20-02947-f006:**
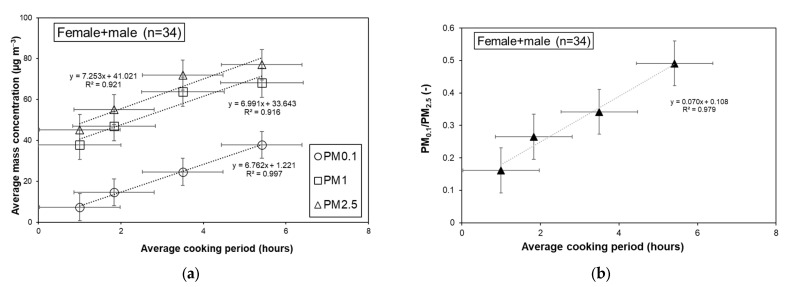
(**a**) Average concentrations of PM plotted against the average cooking period for male and female children; (**b**) the PM_0.1_/PM_2.5_ ratios against the cooking period.

**Figure 7 ijerph-20-02947-f007:**
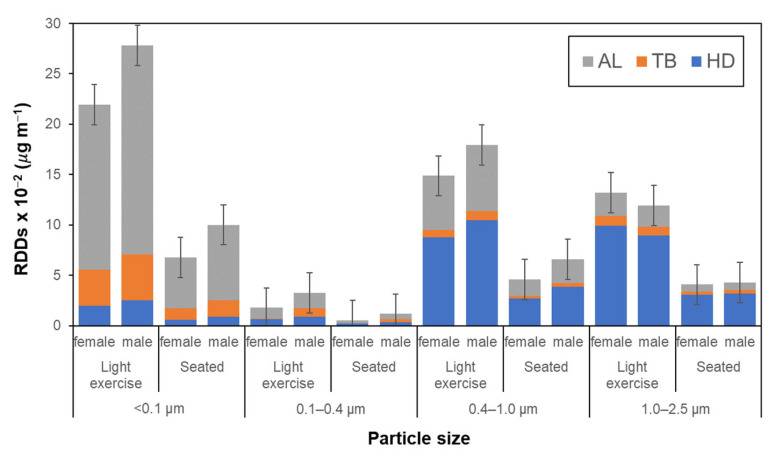
Respiratory deposition doses (RDDs) of PM_1.0–2.5_, PM_0.4–1.0_, PM_0.1–0.4_, and PM_<0.1_ particles in headways (HD), tracheobronchial (TB), and alveoli (AL) regions during light exercise and seated between female and male schoolchildren.

**Table 1 ijerph-20-02947-t001:** Description of selected schools and ambient sampling sites in school environments.

Type		Total Area (m^2^)	Site Sampling	Description
U1	Urban	5537	SG = 2.8 m to RS	Surrounded by hospitals and bank offices, with 71 teachers, 392 male students, and 482 female students.
			SY = 41.1 m to SG	
U2	Urban	4819	SG = 1.3 m to RS	Located in the first central business district (CBD) area of Jambi city in a shopping center, with 57 teachers, 360 male students, and 386 female students.
			SY = 22.1 m to SG	
U3	Urban	4816	SG = 6.5 m to RS	Located near to a former bus terminal which was not in operation, with 89 teachers, 505 male students, and 569 female students.
			SY = 52.3 m to SG	
SU1	Suburban	8800	SG = 4.9 m to RS	Situated the same as SU2, with 54 teachers, 380 male students, and 396 female students. The roadside was crowded with light vehicles (LV), private and mass vehicles, as the inter-provincial highway, but buses and trucks did not pass in this way.
			SY = 35.3 m to SG	
SU2	Suburban	13,141	SG = 3.0 m to RS	Located near a roadside used as people and logistic goods transportation to other provinces, with 52 teachers, 358 male students, and 337 female students.
			SY = 51.6 m to SG	

**Table 2 ijerph-20-02947-t002:** Daily mean (standard deviation) and minimum–maximum of mass-size segregated particles at school gate (SG) and schoolyard (SY).

School		<0.1 (PM_0.1_)	0.1–0.5	0.5–1.0	1.0–2.5	2.5–10	>10
		µg m^−3^	µg m^−3^	µg m^−3^	µg m^−3^	µg m^−3^	µg m^−3^
U1	SG	3.2 ± 1.6	4.8 ± 0.6	4.5 ± 1.9	3.8 ± 1.2	7.3 ± 0.8	2.7 ± 0.4
		(1.6–4.8)	(4.2–5.3)	(2.6–6.4)	(2.5–5.0)	(6.5–8.1)	(2.3–3.0)
	SY	3.0 ± 0.5	3.1 ± 0.7	4.4 ± 0.9	4.5 ± 0.8	11.4 ± 0.7	1.5 ± 1.3
		(2.5–3.4)	(4.2–5.3)	(3.5–5.3)	(3.7–5.2)	(10.7–12.1)	(0.2–2.8)
U2	SG	5.6 ± 2.9	5.3 ± 1.1	5.6 ± 3.3	8.3 ± 1.5	14.4 ± 4.9	7.4 ± 1.8
		(2.7–8.5)	(4.3–6.4)	(2.4–8.9)	(6.8–9.8)	(9.6–19.3)	(5.7–9.2)
	SY	4.3 ± 1.5	3.3 ± 1.3	5.1 ± 0.7	5.9 ± 0.8	12.1 ± 1.9	2.5 ± 1.2
		(2.8–5.8)	(2.0–4.6)	(4.3–5.8)	(5.0–6.7)	(10.1–14.0)	(1.3–3.7)
U3	SG	6.6 ± 1.0	5.5 ± 2.0	9.6 ± 2.2	9.6 ± 2.3	16.7 ± 5.7	8.0 ± 2.3
		(5.6–7.6)	(3.5–7.5)	(7.4–11.8)	(7.2–11.9)	(11.0–22.4)	(5.6–10.3)
	SY	5.8 ± 1.9	4.5 ± 1.8	8.2 ± 2.2	7.0 ± 1.8	13.1 ± 3.8	4.3 ± 1.0
		(4.0–7.7)	(2.7–6.4)	(6.0–10.3)	(5.1–8.8)	(9.4–16.9)	(4.0–7.3)
SU1	SG	9.7 ± 4.9	7.9 ± 1.8	13.3 ± 3.4	14.4 ± 5.1	25.8 ± 12.4	13.9 ± 5.3
		(4.8–14.6)	(6.0–9.7)	(9.8–16.7)	(9.3–19.4)	(13.4–38.2)	(8.5–19.2)
	SY	5.8 ± 0.2	5.2 ± 1.0	10.4 ± 1.8	9.1 ± 1.8	14.6 ± 4.7	5.6 ± 1.7
		(5.6–6.0)	(4.2–6.2)	(8.7–12.2)	(7.2–10.9)	(9.9–19.3)	(4.0–7.3)
SU2	SG	13.8 ± 2.9	11.7 ± 0.5	10.7 ± 3.0	14.8 ± 3.5	35.4 ± 4.6	20.8 ± 4.2
		(10.9–16.8)	(11.2–12.2)	(7.7–13.7)	(11.2–18.3)	(30.8–40.1)	(16.5–25.0)
	SY	2.4 ± 1.6	6.6 ± 1.8	7.2 ± 2.3	4.2 ± 2.8	8.1 ± 2.6	1.0 ± 0.6
		(0.8–4.0)	(4.8–8.5)	(4.9–9.5)	(1.4–7.1)	(5.5–10.7)	(0.4–1.6)

**Table 3 ijerph-20-02947-t003:** Comparison of PM mass concentration related to school environments and roadsides in the present study and previous studies.

Location	Description	Size-Resolved Particles (µg·m^−3^)				References
		PM_0.1_	PM_1.0_	PM_2.5_	PM_10_	
Jambi, Indonesia	School gate/roadside (urban, traffic)	3.3–7.0	11.6–22.2	17.4–30.8	27.5–46.3	This study
	School gate/roadside (suburban, traffic)	7.9–15.7	26.4–42.7	37.4–62.0	60.3–101.8	
	Schoolyard (urban, traffic)	3.1–5.6	11.0–16.7	15.8–23.5	26.5–37.1	
	Schoolyard (suburban, traffic)	3.2–5.0	15.7–23.1	20.6–32.4	29.4–47.6	
North Sumatra, Indonesia	Roadside (urban, traffic)	9.3–17.0	35.7–52.3	48.8–74.5	62.6–99.9	Putri et al. (2021) [63]
	Schoolyard (urban, traffic)	14.3–17.5	49.8–60.7	72.1–90.4	96.5–121	
Jambi, Indonesia	Roadside (urban, traffic)	6.7–14.1	24.5–39.7	30.4–58.9	40.3–84.2	Amin et al. (2021) [36]
Hanoi, Vietnam	Schoolyard (urban, traffic)	8.0–20.7	41.7–128	56.7–198	69.3–246	Tran et al. (2020) [64]

**Table 4 ijerph-20-02947-t004:** Results of statistical tests of personal exposure to mass size-segregated particles to schoolchildren participants with a significance level α = 0.05.

		Size-Fractions	F *	F_critical_ *	*p*-Value	H	U
Personal exposure comparison between schoolchildren from all schools	ANOVA	PM_<0.1_	2.8	2.7	0.0	-	-
		PM_0.1–0.4_	3.0	2.7	0.0	-	-
		PM_0.4–1.0_	0.6	2.7	0.6	-	-
	Kruskal–Wallis	PM_1.0–2.5_ **	-	-	6.0	8.4	-
Personal exposure comparison between schoolchildren in urban and suburban areas	ANOVA	PM_<0.1_	10.6	4.2	0.0	-	-
		PM_0.1–0.4_	1.3	4.2	0.3	-	-
		PM_0.4–1.0_	0.9	4.2	0.4	-	-
	Mann–Whitney	PM_1.0–2.5_ ***	-	-	0.1	-	83.9

* If F > F_critical_, there are significant differences in personal exposure in comparison between all schools or between schoolchildren in urban and suburban areas for ANOVA analysis. ** The Kruskal–Wallis test found no significant difference of personal exposure to PM_1.0–2.5_ due to schoolchildren’s school. *** The Mann–Whitney test found no significant difference of personal exposure to PM_1.0–2.5_ between schoolchildren in urban and suburban areas.

**Table 5 ijerph-20-02947-t005:** Results of a post hoc test for the personal exposure of schoolchildren participants to PM_<0.1_ and PM_0.1–0.4_ with a significance level using α_Bonferroni-corrected_ = 0.005 *.

PM_<0.1_		GROUPS	PM_0.1–0.4_	
*p*-Value (*t* Test)	Significant?		*p*-Value (*t* Test)	Significant?
0.211	No	U1 vs. U2	0.656	No
0.109	No	U1 vs. U3	0.657	No
0.729	No	U1 vs. SU1	0.214	No
0.478	No	U1 vs. SU2	0.325	No
0.353	No	U2 vs. U3	0.846	No
0.036	No	U2 vs. SU1	0.029	No
0.03	No	U2 vs. SU2	0.25	No
0.003	Yes **	U3 vs. SU1	0.06	No
0.015	No	U3 vs. SU2	0.482	No
0.156	No	SU1 vs. SU2	0.004	Yes ***

* The α_Bonferroni-corrected_ (0.005) is a significant level α (0.05) divided by the amount of compared combination that can be made from the test (five situations can be 10 compared combinations: 0.05/10 = 0.005). ** Personal exposure to PM_<0.1_ between schoolchildren from U3 and SU1 (*p*-value (*t* test) < α_Bonferroni-corrected_ (0.005)) was found as the most significantly different. *** Personal exposure to PM_0.1–0.4_ between schoolchildren of U3 and SU1 (*p*-value (*t* test) < α_Bonferroni-corrected_ (0.005)) was found as the most significantly different.

**Table 6 ijerph-20-02947-t006:** Average mass concentrations of PMs: personal exposure vs. school environments.

	Urban			Suburban		
	PM_0.1_	PM_1_	PM_2.5_	PM_0.1_	PM_1_	PM_2.5_
	Mass Concentration (µg m^−3^)					
Personal exposure (PE)	21.0 ± 3.6	55.6 ± 7.7	62.3 ± 7.4	32.8 ± 3.3	65.4 ± 1.0	75.4 ± 1.3
Schoolyard (SY)	4.4 ± 1.2	13.9 ± 3.5	19.6 ± 4.4	4.3 ± 1.5	19.4 ± 2.0	26.5 ± 4.0
School gate (SG)	3.2 ± 1.7	16.9 ± 3.7	20.8 ± 7.4	12.2 ± 2.5	34.5 ± 2.0	49.7 ± 4.5
PE/SY	4.8	4	3.2	7.6	3.4	2.8
PE/SG	4.9	3.3	3	2.7	1.9	1.5

**Table 7 ijerph-20-02947-t007:** Personal exposure to size-resolved airborne particles by the characteristics of schoolchildren.

Description			PM_0.1_	PM_1_	PM_2.5_
			(µg m^−3^)		
Smoking status at home	Urban	Yes	21.4 ± 9.8	58.5 ± 19.4	64.7 ± 19.9
		No	19.3 ± 10.0	50.2 ± 17.2	58.7 ± 18.1
	Suburban	Yes	33.1 ± 13.5	68.6 ± 18.7	80.3 ± 19.9
		No	31.0 ± 3.7	56.7 ± 10.6	62.0 ± 11.9
	All	Yes	27.3 ± 12.9	63.5 ± 19.4	72.5 ± 21.0
		No	25.2 ± 10.0	53.5 ± 14.9	60.3 ± 15.6
Cooking time while personal sampling	≤5 h	n = 12	37.8 ± 10.5	68.2 ± 14.4	77.0 ± 18.1
	3–4 h	n = 12	24.6 ± 2.7	63.7 ± 15.6	72.0 ± 16.5
	1–2 h	n = 8	14.6 ± 2.2	47 ± 20.2	55.1 ± 19.9
	≥1 h	n = 2	7.3	37.8	45.2
House occupants (people)	Urban	3–4	18.8 ± 7.0	57.6 ± 16.7	66.3 ± 16.4
		5–8	23.7 ± 13.2	51.8 ± 22.3	55.9 ± 22.5
	Suburban	3–4	31.6 ± 11.9	60.4 ± 17.0	70.7 ± 19.7
		5–8	35.1 ± 12.0	79.2 ± 10.9	88.6 ± 13.7
	All	3–4	25.2 ± 11.5	59.0 ± 16.5	68.5 ± 17.8
		5–8	29.4 ± 13.4	65.5 ± 22.9	72.2 ± 25.1

**Table 8 ijerph-20-02947-t008:** Descriptive characteristics while sitting or during light exercises and Mean-difference (%) in RDD between the male and female students for different PM size stages. The mean-difference value for RDD is calculated using male student’s RDD as a reference.

	Position	Student	HD	TB	AL	Total RDD ×10^−2^	% of Difference in RDD
			×10^−2^	×10^−2^	×10^−2^		
			(µg m^−1^)	(µg m^−1^)	(µg m^−1^)	(µg m^−1^)	
PM_<0.1_	Light exercise	Female	2.0 ± 1.0	3.6 ± 1.8	16.4 ± 8.0	21.9 ± 10.7	5.90%
		Male	2.5 ± 0.9	4.6 ± 1.7	20.7 ± 7.8	27.8 ± 10.4	
	Seated	Female	0.6 ± 0.3	1.1 ± 0.5	5.1 ± 2.5	6.8 ± 3.3	2.20%
		Male	0.9 ± 0.3	1.6 ± 0.6	7.5 ± 2.8	10.0 ± 3.8	
PM_0.1–0.4_	Light exercise	Female	0.6 ± 0.5	0.1 ± 0.1	1.0 ± 0.8	1.8 ± 1.3	1.50%
		Male	0.9 ± 0.5	0.9 ± 0.5	1.5 ± 0.8	3.2 ± 1.8	
	Seated	Female	0.2 ± 0.1	0.0 ± 0.0	0.3 ± 0.2	0.5 ± 0.4	0.60%
		Male	0.3 ± 0.2	0.3 ± 0.2	0.5 ± 0.3	1.2 ± 0.6	
PM_0.4–1.0_	Light exercise	Female	8.8 ± 4.9	0.7 ± 0.4	5.4 ± 3.0	14.9 ± 8.3	3.00%
		Male	10.5 ± 5.6	0.9 ± 0.5	6.6 ± 3.7	17.9 ± 9.8	
	Seated	Female	2.7 ± 1.5	0.2 ± 0.1	1.7 ± 0.9	4.6 ± 2.6	2.00%
		Male	3.9 ± 2.2	0.3 ± 0.2	2.4 ± 1.3	6.6 ± 3.7	
PM_1.0–2.5_	Light exercise	Female	9.9 ± 6.7	1.0 ± 0.7	2.3 ± 1.6	13.2 ± 8.9	−1.30%
		Male	9.0 ± 6.2	0.9 ± 0.6	2.1 ± 1.4	11.9 ± 8.2	
	Seated	Female	3.1 ± 2.1	0.3 ± 0.2	0.7 ± 0.5	4.1 ± 2.8	0.20%
		Male	3.2 ± 2.2	0.3 ± 0.2	0.7 ± 0.5	4.3 ± 3.0	

## Data Availability

Not applicable.

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
