# Peer review of "Investigation of the Exposure of Schoolchildren to Ultrafine Particles (PM0.1) during the COVID-19 Pandemic in a Medium-Sized City in Indonesia"

_ijerph, 2023, doi:10.3390/ijerph20042947_

Round 1

Reviewer 1 Report

1. The introduction needs to be further revised to highlight the purpose of the study, You need to introduce what others have studied and what needs further research. Besides, the following all of references are recommended to be cited:
https://www.sciencedirect.com/science/article/pii/S0167732221011296 2. The relevance/novelty of the work needs to be highlighted.
3. Check the grammar throughout the article and correct it. Proofread the article as many language errors were identified.

Author Response

Dear reviewer,

Thanks for your comments and suggestions. Please see the attachment for point to point responses of the comments and suggestions. 

Thank you 

Reviewer 2 Report

Thank you for this interesting paper

Major points

there are no health outcomes other than respiratory symptoms, do you have data on lung function?

You have not reported any student specific internal dose of exposure - how clear are you that the calculations you have done accurately represent how much the child breaths in?

There is a lot of comparisons - they have been adjusted for - but I am unclear  why each school was compared with the other

Author Response

Dear reviewer,

We thank reviewer for the appreciation of our manuscript. Regarding your comments and suggestions, please see the attachment for point to point responses of the comments and suggestions.

Thank you.

Reviewer 3 Report

The article ”Investigation of the exposure of schoolchildren to ultrafine particles (PM0.1) during the COVID-19 pandemic in a medium- sized city in Indonesia”

intends to analyses the health risk for schoolchildren exposed airborne fine and ultrafine particles (PM0.1) and schoolchildren’s behaviors in their daily life during a period of the COVID-19 pandemic in five public schools Jambi City. It was used an inertial filter technology for the air sampling, Ambient Nano-Sampler. The Personal Nano-Samplera was applied for the evaluation of PM0.1 exposure.

The statistical analysis was applied to dataset consisting in a Parametric and non-parametric determination. Also the ANOVA test was used for the analysis of personal exposure to PM0.1, PM0.1–0.4 and PM0.4–1

This article is clearly prepared, the information is well presented and can be applied in the analysis of different types of educational spaces. Since the article analyzes the exposure of school children to air pollution, I believe that this article deserves to be published.

This paper has a major main contribution in the literature by presenting clearly and gradually the way in which the determinations and interpretations of the results were performed.

Author Response

Dear reviewer,

We thank reviewer for the appreciation of our manuscript. 

Thank you

Reviewer 4 Report

This is a very interesting study evaluating the personal exposure of school children to different size fractions of PM. The authors find that personal exposure to indoor PM was quite higher (1.5 to 7.6x) than to outdoor PM, particularly for the smallest size fraction (ultrafine). A particular focus on cooking emissions contributing to these higher exposures in indoor environments is a strong novelty and finding of this study.

The authors mention the particularly higher health concern involved with UFPs penetrating deeper into the alveolar region of the lungs. This is a very interesting point, and I think the authors should cite a few studies that have directly measured the effect of lung-deposited surface area (LDSA) of particles. LDSA is a more direct indicator of the alveolar toxicity mentioned by the authors, and thus would be appropriate to mention here. I suggest citing the following studies:

1. Maier, K. L., Alessandrini, F., Beck-Speier, I., Josef Hofer, T. P., Diabaté, S., Bitterle, E., ... & Schulz, H. (2008). Health effects of ambient particulate matter—biological mechanisms and inflammatory responses to in vitro and in vivo particle exposures. Inhalation toxicology, 20(3), 319-337.

2. Donaldson, K., Tran, L., Jimenez, L. A., Duffin, R., Newby, D. E., Mills, N., ... & Stone, V. (2005). Combustion-derived nanoparticles: a review of their toxicology following inhalation exposure. Particle and fibre toxicology, 2(1), 1-14.

3. Oberdürster, G. (2000). Toxicology of ultrafine particles: in vivo studies. Philosophical Transactions of the Royal Society of London. Series A: Mathematical, Physical and Engineering Sciences, 358(1775), 2719-2740.

Author Response

Dear reviewer,

We thanks reviewer for the appreciation of our manuscript and we considered the suggestions. Please see the attachment for detail of responses from your suggestion.

Thank you

Round 2

Reviewer 2 Report

Thank you for the revisions - I am now happy with the standard